# An outbreak of pulmonary tuberculosis and a follow-up investigation of latent tuberculosis in a high school in an eastern city in China, 2016–2019

**Yirong Fang[1][☉], Yan Ma[1][☉], Qiaoling Lu[1], Jiamei Sun[1], Yingxin Pei[2]\***

**1** Department of Infectious Disease, Shaoxing Center for Disease Control and Prevention, Shaoxing City, Zhejiang Province, China, **2** Chinese Field Epidemiology Training Program, Office of Education and Training, Chinese Center for Disease Control and Prevention, Beijing, China

☉ These authors contributed equally to this work.
* peiyx@chinacdc.cn

**Data Availability Statement:** All relevant data are within the manuscript and its Supporting Information files.

## Abstract

### Background

In October 2016, a senior high school student was diagnosed with sputum-smear positive [SS(+)] pulmonary tuberculosis (TB). We conducted an investigation of an outbreak in the school, including among students and teachers diagnosed with latent TB, who we followed until July 2019.

### Methods

We defined latent TB infection (LTBI) as a tuberculin skin test (TST) induration of 15mm or larger; probable TB as a chest radiograph indicative of TB plus productive cough/hemoptysis for at least 2 weeks, or TST induration of 15mm or larger; and confirmed TB as two or more positive sputum smears or one positive sputum smear plus a chest radiograph indicative of TB or culture positive with *M. tuberculosis*. We conducted mycobacterial interspersed repetitive unit–variable number tandem repeat (MIRU-VNTR) typing based on 24 loci in the isolates.

### Results

Between October 2016 and July 2019, we identified 52 cases, including nine probable, six confirmed, and 37 LTBI cases. The index case-student had attended school continuously despite having TB symptoms for almost three months before being diagnosed with TB. We obtained three isolates from classmates of the index case in 2016; all had identical MIRU-VNTR alleles with the index case. The LTBI rate was lower among students (7.41%, 30/405) than among teachers (26.92%, 7/26) (rate ratio [RR] = 0.28, 95% confidential interval [CI]: 0.13–0.57). Among the 17 students who had latent TB and refused prophylaxis in October 2016, 23.53% (4/17) became probable/confirmed cases by July 2019. None of the six teachers who also refused prophylaxis became probable or confirmed cases. Of the 176

**Funding:** Our research was funded by Chinese Field Epidemiology Training Program, Chinese Center for Disease Control and Prevention; the grant number is 131031001000160016.

**Competing interests:** The authors have declared that no competing interests exist.

students who were TST(-) in October 2016, 1.70% (3/176) became probable/confirmed cases, and among the 20 teachers who were TST(-), 1 became a probable case.

## Conclusions

Delayed diagnosis of TB in the index patient may have contributed to the start of this outbreak; lack of post-exposure chemoprophylaxis facilitated spread of the outbreak. Post-exposure prophylaxis is strongly recommended for all TST-positive students; TST-negative students exposed to an SS(+) case should be followed up regularly so that prophylaxis can be started if LTBI is detected.

## Introduction

China is one of 30 countries designated by the World Health Organization (WHO) as having a high burden of tuberculosis (TB) during the period 2016 through 2020 [1]. Tuberculosis epidemics in China have been characterized as having large numbers of infected people, with many symptomatic patients, deaths, rural patients, and patients with drug-resistant TB [2]. In China, tuberculosis outbreaks often occur in educational settings such as kindergartens, primary schools, high schools, and universities [3]. In 2018, 48,289 students were reported to have pulmonary TB, an incidence of 17.97/100 000 [4]. Crowded dormitories and close proximity in classrooms can facilitate transmission of TB, and known risk factors for TB outbreaks in schools are close contact, inadequate ventilation, and delayed diagnosis [5–12]. However, few studies determine outcomes of latent TB infections.

In October 2016, a student in a senior high school with 8 classes and 405 students in eastern China's Zhejiang Province was diagnosed with sputum-smear positive [SS(+)] pulmonary TB. We investigated this school-based outbreak to identify individuals with latent TB who would benefit from prophylactic treatment and prevent further transmission of TB. After offering treatment, we followed outcomes of individuals with latent TB for their three remaining years of high school. We report results of our investigation and follow-up, and we provide recommendations for control and prevention of TB outbreaks in schools.

## Methods

### Study design

We reviewed medical records and interviewed physicians and nurses who worked in the school's health clinic. We conducted in-person interviews of case-students or telephoned their parents to obtain information regarding disease onset and exposure history. We reviewed administrative records to evaluate measures taken by Shaoxing Center for Disease Control and Prevention (Shaoxing CDC) to control the outbreak.

### Tuberculin skin test procedures

Shaoxing CDC staff performed tuberculin skin tests (TST) using standard guidelines. A qualified nurse used intradermal injections on the medial left forearm to administer 0.1ml (2 IUs) of Purified Protein Derivative (PPD) produced from BCG (Chengdu Institute of Biological Products, Chengdu, China). After 72 hours, a qualified physician measured the transverse induration (in mm) at the TST site [13,14]. A strong Mantoux positive result was defined as a TST induration of 15 mm or larger, and/or blisters, necrosis, and lymphangitis [13].

## Case definition

We defined latent TB infection (LTBI) as a TST induration ≥15mm in diameter in a student or teacher of the high school [13]; probable TB as a chest radiography indicative of TB, plus at least one of the following: productive cough or hemoptysis lasting for ≥2 weeks, or TST≥15 mm; and confirmed TB as two or more positive sputum smears or one positive sputum smear plus a chest radiograph indicative of TB or culture positive with *M. tuberculosis*.

## Case finding

We used self-administered questionnaires to screen students and teachers for symptoms (i.e., productive cough or night sweats lasting for ≥2 weeks). Individuals with TSTs ≥15mm or TB symptoms were also screened by chest radiography [13]. If chest radiography was inconclusive, a computer tomography (CT) scan was performed [13]. For individuals with abnormal chest radiographs, CT scan findings, or symptoms indicative of TB, three unconcentrated sputum specimens (night, morning, and spot samples) were examined by microscopy and cultured [14].

## MIRU-VNTR-typing

We performed mycobacterial interspersed repetitive variable numbers of tandem repeats analyses (MIRU-VNTR) on *Mycobacterium tuberculosis* (*M. tuberculosis*) isolates to determine genetic relationships among isolates, with 24 loci: MIRU4, MIRU26, MIRU40, MIRU10, MIRU16, MIRU31, VNTR42-MTUB04, NTR43-ETR-C, ETR-A, VNTR47-MTUB30, VNTR52-MTUB39, VNTR53-QUB4156, QUB-11b, VNTR 1955-MTUB21, QUB-26, MIRU02, MIRU 23, MIRU39, MIRU20, MIRU24, MIRU27, VNTR46-MTUB29, VNTR48-ETRB, and VNTR49-MTUB34.

We performed MIRU-VNTR typing with methods as previously reported [15–17]. Polymerase chain reaction (PCR) fragments were analyzed using 1.5% agarose gel electrophoresis with a 100-bp DNA ladder as the molecular weight standard. The number of tandem repeats was based on the length of the repeat and flank sequences for each locus. H37Rv PCR products were loaded to ensure accuracy, and PCR products from sterile water were used to control for reagent contamination. Minimum spanning trees were constructed to show genetic relationships among the isolates in the outbreak and endemic strains in Zhejiang province collected that were obtained in 2007 [18]. We used MIRU-VNTRplus to construct the minimal spanning tree (https://www.miru-vntrplus.org/MIRU/treeBatch.faces). Drug-susceptibility testing was performed with MIRU-VNTR typing and the liquid rapid drug sensitivity method with the following four tuberculosis drugs: isoniazid, streptomycin, rifampicin and ethambutol.

## Statistical analysis

Statistical analyses were performed using the SPSS statistical package (version 11.0). Rate Ratio and 95% confidential interval were used to compare the incidence difference of different population.

## Ethical approval and consent

This investigation was conducted in response to a public health emergency, and was therefore exempt from ethical review in accordance with China's Regulations on Emergency Response to Public Health Emergencies (http://www.gov.cn/banshi/2005-08/02/content_19152.htm). Data were de-identified to protect confidentiality. Written consent was obtained from all

participants prior to interview. For students, written, informed consent was obtained from their parents.

## Results

### Epidemiological investigation

In 2016, the senior high school had eight first-year classes; there were 405 students—204 males and 201 females. There were approximately 50 students in each class and between 10–12 students in each dormitory.

Our investigation identified nine probable cases and six confirmed cases of TB. The index case-student started to develop symptoms (productive cough and fever) on April 20, 2016. She sought medical care and was diagnosed with bacterial pneumonia; she received antibiotics for treatment. Her cough persisted while other symptoms resolved. After graduation from junior high school in June 2016, she attended a pre-matriculation course for incoming senior high school students on July 11–30 and August 15–20—prior to school opening on September 8. Her cough worsened on September 12 and she went to Hospital A where she was diagnosed with TB with SS(+) the next day. She continually attended school during the 2 months between the pre-matriculation course and confirmation of her SS(+) TB diagnosis; she was hospitalized and excluded from school after being diagnosed. The investigation determined that her father had been diagnosed with SS(+) TB in May 2014.

On October 21 2016, Shaoxing CDC began TST screening of students (n = 52) and teachers (n = 6) who shared a classroom with the index case-student. After an additional four students were diagnosed with TB from the same classroom, on October 30, Shaoxing CDC expanded screening to students (n = 141) and teachers (n = 20) who were in or taught the same grade with the index case-student. We found that 8.81% (17/193) of students and 23.08% (6/26) of teachers had latent TB infection (TST$\geq$15mm); three students were diagnosed as confirmed cases and two students were diagnosed as probable cases. Among these five, four shared the same classroom with the index case-student, another had shared a classroom with the index case-student in junior high school from September 2014 to June 2016. The junior high school classmate was identified as a TB case during screening in senior high school. No other suspected TB cases were found in the junior high school based on the information provided by teachers. No screening was conducted of other classmates in the junior high school because these students matriculated at different senior high schools. None of her junior high school classmates were among the reported TB cases in same administrative area during the 2016–2018 study period. The index case's dormitory roommates were all screened because they were also the index case's classmates.

Free isoniazid prophylaxis was provided for students and teachers with latent TB infection. However, all 17 students and 6 teachers refused prophylactic treatment, reportedly due to fear of potential hepatotoxicity from isoniazid.

On March 29 2018, one student in the same classroom with the index case-student was diagnosed with SS(-) TB after exhibiting TB symptoms and seeking medical care. On April 4 2018, another student in a neighboring classroom was diagnosed with SS(+) TB after exhibiting TB symptoms and seeking medical care. Both were in the same grade and were TST-positive in the 2016 TST screening but had refused prophylaxis. Subsequently, Shaoxing CDC conducted another round of TST screening among the 253 students in the grade and the 26 teachers who taught at this grade level. In that screening, 22 students (nine of whom had been TST-positive in the 2016 TST screening) and one teacher was found to have latent TB infection; three additional students sharing a classroom with the primary case-student were identified with SS(-) TB.

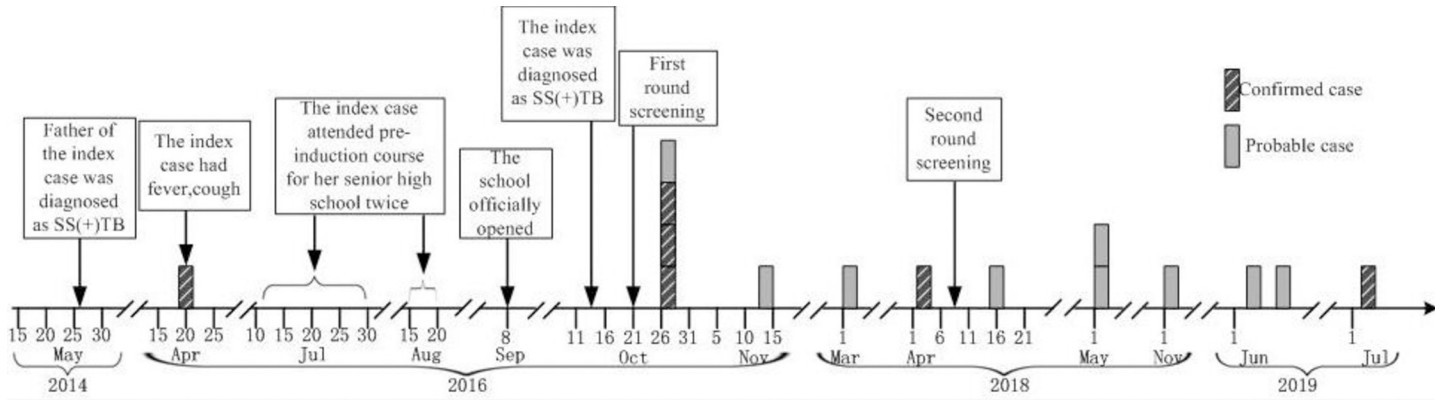

**Abbreviations**:   SS(+): sputum-smear positive; TB: tuberculosis

**Fig 1. Timeline of probable and confirmed TB cases in a high school: Zhejiang Province, China, 2016 to 2019.**

Shaoxing CDC implemented mandatory isolation and treatment of all probable and confirmed case-students identified during the second round of screening. Free prophylactic treatment was offered to students and teachers found to have latent TB infection. Among those offered prophylaxis, 17.39% (4/23) completed the recommended 6-month regimen, and none developed TB during the follow-up period.

Shaoxing CDC monitored affected students and teachers until the students graduated from high school on July 2019. One student was diagnosed as SS(-) TB disease in November 2018 and two students that shared a classroom with the index case-student were diagnosed with SS (-) TB on June 13 and 24. One teacher was diagnosed with SS(+)TB on July 12, 2019 (Fig 1).

Between October 2016 and July 2019, we identified a total of 52 cases (nine probable, six confirmed TB, and 37 LTBI). The LTBI rate was lower among students (7.41%) than among teachers (26.92%) (rate ratio [RR] = 0.28, 95% confidence interval [CI]: 0.13–0.57); 3.46% of the students and 3.85% of the teachers developed probable/confirmed TB during the 3 years of follow up (Table 1).

## Risk factor analysis

Among the 405 students in the same grade as the index case-student, sharing a classroom was significantly associated with higher risk of probable/confirmed TB (RR = 37.37, 95% CI: 8.51–163.73). The attack rate among students in the same dormitory as the index case was significantly higher than among students in different dormitories (RR = 8.71, 95% CI: 3.05–24.88).

**Table 1. Latent infection and probable/confirmed TB among students in the same grade with index SS(+) case-student and their teachers in a high school—Zhejiang Province, China, 2016–2019.**

| Group | N | Latent TB Infection | | | Probable/Confirmed Cases | | |
|---|---|---|---|---|---|---|---|
| | | n | Rate (%) | RR (95% CI) | n | Rate (%) | RR (95% CI) |
| Students | 405 | 30[a] | 7.41 | 0.28 (0.13–0.57) | 14 | 3.46 | 0.90 (0.12–6.57) |
| Teachers | 26 | 7[b] | 26.92 | | 1 | 3.85 | |

[a] Including 17 identified in October 2016 and 13 identified in May 2018.

[b] Including 6 identified in October 2016 and 1 identified in May 2018.

Abbreviations: RR, Rate Ratio; CI, Confidence Interval.

**Table 2. Probable/confirmed TB cases among students in the same grade with index SS(+) case, by exposures: Zhejiang Province, China, 2016–2019[a].**

| Exposure | N | Probable/Confirmed Cases | | RR (95% CI) |
|---|---|---|---|---|
| | | n | Rate (%) | |
| Gender | | | | |
| Male | 204 | 5 | 2.45 | 0.62 (0.20–1.85) |
| Female | 201 | 8 | 3.98 | Ref |
| Same classroom as index case | | | | |
| Yes | 52 | 11 | 21.16 | 37.37 (8.51–163.73) |
| No | 353 | 2 | 0.57 | Ref |
| Same dormitory as index case | | | | |
| Yes | 12 | 4 | 33.33 | 8.71 (3.05–24.88) |
| No | 209 | 8[b] | 3.83 | Ref |

[a] Excluding the index case-student.

[b] One student-case was not considered as living in the dormitory because the student commuted from home to school every day and did not live in the dormitory.

Rates of latent TB and probable/confirmed TB did not differ significantly by sex (RR = 0.62, 95% CI: 0.20–0.1.85) (Table 2).

## Follow up

During October 2016, 193 students and 26 teachers were screened with TSTs. Shaoxing CDC monitored the students and teachers until the students graduated from high school in July 2019. Of the 17 students who had latent TB infection in October 2016, 23.53% (4/17) became probable/confirmed cases. Of the six teachers who had latent TB in October 2016, none became probable or confirmed cases. Of the 176 students who were TST(-) in October 2016, 1.70% (3/176) became probable/confirmed cases. Of the 20 teachers who were TST(-) in October 2016, one became a confirmed case (Table 3).

## MIRU-VNTR-typing

Shaoxing CDC obtained 15 sputum specimens from suspected cases and the index case and performed sputum smears; four were positive, including two specimens positive with one from the index case in 2016 and two specimens positive in subsequent years. *M. tuberculosis* was isolated from six of the 15 sputum specimens with four strains isolated in 2016 including index case and two strains isolated in subsequent years. MIRU-VNTR results from the isolates in 2016 demonstrated that these four strains contained the same MIRU-VNTR alleles in 24 loci and minimum spanning tree, indicating that they belong to the Beijing family based on

**Table 3. Following up results of 193 students and 26 teachers in the same grade as the index case-student: Zhejiang Province, China, 2016–2019[a].**

| | Screening results in October 2016 | N screened | Following up results from October 2016 to July 2019 | | RR (95%CI) |
|---|---|---|---|---|---|
| | | | number of Probable/Confirmed Cases | Attack Rate (%) | |
| students | TST-positive (≥15mm) | 17 | 4 | 23.53 | 14.04 (3.42–57.61) |
| | TST-negative (<15mm) | 176 | 3 | 1.70 | Ref |
| teachers | TST-positive (≥15mm) | 6 | 0 | - | - |
| | TST-negative (<15mm) | 20 | 1 | 5.0 | - |

[a] Excluding a case who was in the same classroom as index case but not screened by TST in 2016.

the international database https://www.miru-vntrplus.org/MIRU/index.faces (Fig 2) [19]. Drug-susceptibility testing was performed with the liquid rapid drug sensitivity method with the following four tuberculosis drugs: isoniazid, streptomycin, rifampicin and ethambutol. All four of the isolates were sensitive to these four drugs.

## Discussion

Our epidemiological investigation and analyses showed that this TB outbreak started with delayed diagnosis and therefore delayed isolation of the index SS(+) case-student. The outbreak was facilitated by long-term contact among students in classrooms and a dormitory. The father of the index case contracted TB two years before the index case, suggesting that this outbreak is an example of TB transmission from community to school. Several latent TB cases converted to probable/confirmed cases due to refusal of chemoprophylaxis.

We used MIRU-VNTR laboratory typing methods in this outbreak investigation to complement findings from our traditional epidemiological investigation and to help determine the source of infection and linkages among cases. These laboratory data supported findings from

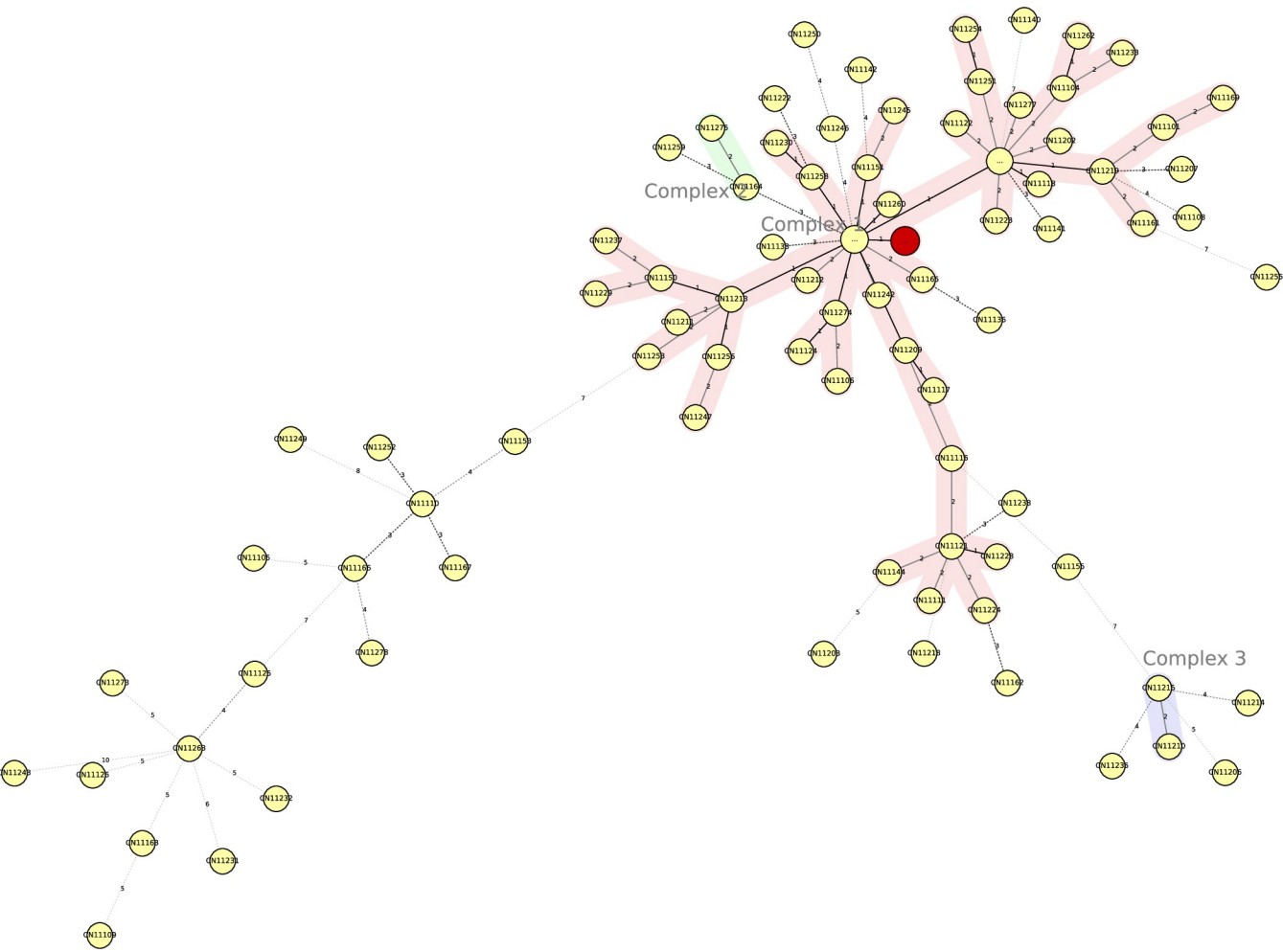

**Fig 2. Minimum spanning tree showing clustering by MIRU-VNTR of four M. tuberculosis isolates from this outbreak.** Each nodal point represents a cluster with identical genotypes, and the sizes of the nodal points are scaled to the number of strains in the cluster. Red shading indicates the outbreak group, which belong to the Beijing family. "Complex" refers to a group of strains with less than three loci variants that are linked with different colors.

the epidemiological investigation. Close contacts were followed up for almost three years allowing outcomes of latent TB infection and close contacts to be observed for progress to TB.

Limited space and poor ventilation in classrooms and dormitories, long-time continuous contact, and delayed diagnosis and treatment TB patients have been shown to be risk factors for TB outbreak in schools [7–12,20]. Therefore, early-detection, early isolation, and early treatment of TB cases and prophylaxis for latent TB infection are important measures to interrupt TB transmission. None of these effective measures was observed in this outbreak. In contrast, the index case was diagnosed as SS(+) TB 6 months after her initial TB-associated symptoms and had continually attended school, causing eleven students with whom she shared a classroom or dormitory to contract TB in the subsequent 3 years. Opportunities for exposure for teachers and students were different, resulting in a lower LTBI rate among students, as the teachers had apparently been more frequently exposed to the index case than did the students. Teachers had a higher LTBI rate than the students did, as LTBI rates are known to increase with age [21].

Low compliance with chemoprophylaxis was seen in other studies [22–24], which is consistent with our finding that all 17 students with latent TB identified in 2016 refused prophylaxis, due either to misunderstanding the necessary of prophylaxis when there are no symptoms or to fear of side effects. As a consequence, 23.53% (4/17) PPD(+) students became probable or confirmed cases compared with 1.70% (3/176) of PPD(-) students, suggesting that individuals exposed to an SS(+) case should be followed up regularly in order to have rapid initiation of prophylaxis once diagnosed with latent TB infection—even if an initial PPD is negative. Consistent with our findings, another study found that about 20% of individuals with latent TB will become TB cases due to lack of prophylaxis. The current standard prophylaxis medicine is isoniazid, which has favorable safety and effectiveness profiles [25,26]. Education is necessary to increase compliance to prevent further transmission in schools. Communication strategies about chemoprophylactic treatment should be adopted, and approaches to increase adherence to regimen should be identified and implemented.

As an adjunct to the investigation, we conducted molecular epidemiological analyses. Although whole-genome sequencing (WGS) is increasingly being used to determine *Mycobacterium tuberculosis* relatedness and is known to deliver greater specificity than MIRU-VNTR, WGS is not always available, and MIRU-VNTR can be used to determine relatedness. MIRU-VNTR has high discriminatory ability and is commonly used to identify clustering of TB cases that have epidemiological links [27–30]. We found three isolates from classmates of the index case and showed that all of the MIRU-VNTR alleles were identical with the index case's —all belonging to the Beijing family, which is the most prominent MTB lineage in East Asia [31,32]. The Beijing genotype is the predominant lineage in Zhejiang, and the distribution of Beijing-genotype strains shows geographic diversity [33]. As the Beijing strain is the endemic strain in Zhejiang, this concordance helps explain the outbreak. Genotyping verified and complemented the epidemiological findings that the index case spread TB to her classmates and her roommates through persistent contact due to her delayed diagnosis and lack of initial isolation.

Our investigation had limitations. First, MIRU-VNTR typing was not conducted among the index case and the other nine cases that were identified after 2016 because sputum culture results were not available for the latter. Second, the father of index case was confirmed with TB in 2014, but without his isolate of *M. tuberculosis* we were unable to confirm through genotyping that the index case was infected by her father. Third, as per national guidelines for prophylaxis, we defined latent TB infection (LTBI) as TST induration ≥15mm in diameter. This cut-off is more conservative than a ≥10mm cut-off, and will result in fewer LTBI cases being shown to progress to TB than would have been shown had a 10 mm cut-off been used.

## Conclusions

School-based tuberculosis outbreak among adolescents should be regarded as high priority and should be prevented. Based on our findings, early diagnosis is paramount to prevent further spread among close contacts in the school. Health care workers in school clinics and health facilities should receive training to ensure their ability to properly diagnose and manage tuberculosis. Second, mechanisms should be established in schools, including morning health-checks, absentee monitoring, entrance medical examinations, and routine medical examination to identify potential TB patients as soon as possible. Third, we should pay attention to health education in schools and raise awareness of teachers and students about tuberculosis so they can seek medical evaluation if they have TB symptoms. Finally, post-exposure prophylaxis is strongly recommended for all TST-positive students; students who are TST-negative and exposed to a SS(+) case should be monitored in order to initiate prophylaxis if they convert to LTBI.

## Supporting information

**S1 File. MIRU-VNTR genotyping results of 24 locus of M. tuberculosis isolates.**
(XLS)

**S2 File. Distribution of different TST diameter.**
(XLSX)

**S3 File. Questionnaire.**
(DOC)

**S4 File. Questionnaire (in Chinese).**
(DOC)

## Acknowledgments

We are grateful to the contribution of Dr. Chen Wei and Dr. Zhou Yang from National Centre for Tuberculosis Control and Prevention, Beijing, China. We appreciate the English language editing by Lance Rodewald, senior advisor at China CDC.

## Author Contributions

**Conceptualization:** Yirong Fang, Yan Ma, Yingxin Pei.

**Data curation:** Yirong Fang, Jiamei Sun.

**Formal analysis:** Yingxin Pei.

**Investigation:** Yirong Fang, Yan Ma, Qiaoling Lu, Jiamei Sun, Yingxin Pei.

**Methodology:** Yan Ma, Qiaoling Lu.

**Resources:** Yirong Fang, Yan Ma.

**Software:** Yirong Fang, Jiamei Sun.

**Supervision:** Yirong Fang.

**Validation:** Yirong Fang, Qiaoling Lu, Yingxin Pei.

**Writing – original draft:** Yan Ma, Yingxin Pei.

**Writing – review & editing:** Yirong Fang, Yan Ma, Yingxin Pei.

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
