## [Decision Letter · Decision Letter 0]

1 Oct 2020

PONE-D-20-26099

An outbreak of pulmonary tuberculosis and follow-up of latent tuberculosis infection in a high school: China, 2016-2019

PLOS ONE

Dear Dr. Pei,

Thank you for submitting your manuscript to PLOS ONE. After careful consideration, we feel that it has merit but does not fully meet PLOS ONE’s publication criteria as it currently stands. Therefore, we invite you to submit a revised version of the manuscript that addresses the points raised during the review process.

Please address all comments of the reviewers regarding study design, methodology and interpretation of your results. Also please note that English language should be corrected by a native speaker, and all typos should be corrected.

We look forward to receiving your revised manuscript.

Kind regards,

Igor Mokrousov, Ph.D., D.Sc.

Academic Editor

PLOS ONE

Journal Requirements:

2. You indicated that ethical approval was not necessary for your study. We understand that the framework for ethical oversight requirements for studies of this type may differ depending on the setting and we would appreciate some further clarification regarding your research. Could you please provide further details on why your study is exempt from the need for approval and confirmation from your institutional review board or research ethics committee (e.g., in the form of a letter or email correspondence) that ethics review was not necessary for this study? Please include a copy of the correspondence as an "Other" file.

Reviewers' comments:

Reviewer's Responses to Questions

**Comments to the Author**

1. Is the manuscript technically sound, and do the data support the conclusions?

Reviewer #1: Yes

Reviewer #2: Yes

Reviewer #3: Partly

Reviewer #4: Yes

2. Has the statistical analysis been performed appropriately and rigorously? 

Reviewer #1: Yes

Reviewer #2: I Don't Know

Reviewer #3: Yes

Reviewer #4: Yes

3. Have the authors made all data underlying the findings in their manuscript fully available?

Reviewer #1: Yes

Reviewer #2: Yes

Reviewer #3: Yes

Reviewer #4: Yes

4. Is the manuscript presented in an intelligible fashion and written in standard English?

Reviewer #1: Yes

Reviewer #2: No

Reviewer #3: No

Reviewer #4: Yes

5. Review Comments to the Author

Reviewer #1: This well-designed epidemiological study presenting the results of an investigation of a school-based tuberculosis outbreak confirms previous findings in similar settings provided by other authors. An appropriate set of statistics tools well controlled the statistical reliability of epidemiologically correct conclusions, which are strengthened by an important follow-up section of the study. The paper is clearly written.

I support publishing this paper.

Specific questions/comments.

1. Low quality of Fig 2 – please, change the format.

2. Please, clarify in the text the total number of cultures and the total number of culture-confirmed patients (only 4?) in your study.

3. Please mention DST methods and at least phenotypic drug-susceptibility profiles of Mt isolates clustered by MIRU-VNTR.

4. MIRU-VNTR clustered digital profiles should be assigned according to international databases: http://www.pasteur-guadeloupe.fr:8081/SITVIT2/query (http://www.pasteur-guadeloupe.fr:8081/SITVIT2/submit.jsp)

and https://www.miru-vntrplus.org/MIRU/index.faces

5. To my mind, the term Homology analysis of isolates does not imply MIRU-VNTR data but rather implies some other tools to compare nucleotide or protein sequences to sequence databases and calculates the statistical significance of matches (https://www.ncbi.nlm.nih.gov/guide/homology/) https://molbiol-tools.ca/Homology.htm

Maybe ‘MIRU-VNTR-typing’ will be more relevant?

6. Line 305 …family, which is the most prominent MTB lineage in East Asia[27],

This reference [27] refers to D. van Zoolingen et al. paper of 1995 and definitely should be preserved in the list. However, there is a significant number of quiet recent publications on Beijing strains circulating in China to be referred to additionally.

For example: Merker M., Blin C., Mona S. et al. Evolutionary history and global spread of the Mycobacterium tuberculosis Beijing lineage. Nat Genet. 2015;47(3):242-9. https://doi.org/10.1038/ng.3195

7. Small corrections:

Remove a string of two hyphen-minus characters (--) in VNTR 48--ETRB (column Y in Table, Supporting information).

Reviewer #2: I could not find any difference between this report and the author's previous report (Outbreak of pulmonary tuberculosis in a Chinese high school, 2009-2010. J Epidemiol 2013;23:307-12), ref. 2 in the bibliography. The settings, findings, and conclusions are identical in both reports. The take-home message of both articles is the same, that is, the importance of early identification of active Tuberculosis cases and the need to treat latent tuberculosis among close contacts of active tuberculosis patients.

The article needs a comprehensive language editing and a more comprehensible data presentation.

Several issues need the author's attention such as:

1. Why did the authors defined TST>15 as positive rather than TST>5mm, which is the standard definition of TST positivity among recent contacts of infectious tuberculosis cases. The authors should also elaborate regarding the screening procedure (and TST results (>5mm, >10mm, >15mm) distribution, particularly because the citation they provide in this regard is written in Chinese (ref 13), which is not accessible to many "PLOS ONE" readers.

2. The number of active cases in table 2 does not match (13 and 12 out of 14).

3. "Tuberculosis" case definition should include having a positive culture result, and "probable tuberculosis" definition should include a clinical improvement and resolution of chest X-ray findings following treatment.

5. The article Title is misleading ":China, 2016-2019"

Reviewer #3: Review

The manuscript " An outbreak of pulmonary tuberculosis and follow-up of latent tuberculosis infection in a high school: China, 2016-2019” by Yingxin Pei describes outbreak investigation and evaluate a possible nature of this outbreak, along with the source, and transmission route.

This paper describes interesting follow-up study. Authors brought some additional evidence for the importance of post-exposure TB prophylaxis and highlighted possible transmission of M. tuberculosis in school settings.

However, to my opinion, there are few issues that should be addressed.

1. Several studies on the outbreak investigations in schools in China have been published recently such as Pan et al., 2018; Xu et al., 2019; You et al., 2019; Hou et al., 2020; Bao et al., 2019. Authors should clearly emphasize the importance/novelty of the current study and findings.

2. English editing English should be corrected by a native speaker, and all typos should be corrected. Just few examples:

Line 122: please change to “night sweats”. Also: “Teacher” should be in plural?

Line 131: Here and elsewhere. Mycobacterium tuberculosis in italics.

Line 130 Please change to “variable numbers of tandem repeats”

Lines 133-137: please correct punctuation and the usage of “-“ and “—“

Line 138: please correct “typing method was” or “typing methods were” .

Methods.

3. What software was used to compute the minimal spanning tree?

4. More detailed information about M. tuberculosis isolates obtained in year 2007 should be provided.

5. Line 145. What do you mean by “epidemic strains”?

6. Some additional data on the population settings of this study should be provided such as persons per room and students per class.

Results

7. Please provide P value for the results, as defined in Materials and Methods.

8. Lines 250-252. Please rephrase for clarity.

9. Lines 252-255. Did you used any of online resources such as reference databases/analysis tools for the analysis of MIRU-VNTR results such as MIRU-VNTRplus?

10. Figure 2 is of insufficient quality (impossible to read) and insufficiently described. How many isolates were used to construct the tree? What genotypes were included? What is the meaning of green color?

11. The authors stayed (lines 77-81) that China is one of 30 high TB burden countries, and TB outbreaks often occur in institutional settings. Genotyping results are available only for 4 cases. Thus, there is a possibility that all other identified probable, confirmed and LTBI cases did not originated from the index case and/or belonged to this outbreak.

12. Line 265. How the ventilation in the classroom and dormitory was assessed? As no evidence of the poor ventilation is provided in the manuscript, this statement is groundless. Please remove or rephrase.

13. Line 300-301. This information is quite outdated and not fully correct. First, at present, whole-genome sequencing (WGS) is widely used to determine Mycobacterium tuberculosis relatedness and is known to deliver greater specificity. Second, the finding in study by David et al., 2018 (Reference 25) showed that, in the setting studied, MIRU-VNTR typing and epidemiological risk factors were poorly predictive of close genomic relatedness, assessed by single nucleotide variants. Also, MIRU-VNTR performance varies markedly by lineage. Please rewrite.

14. Line 304-307. In my opinion, the fact that Beijing family is the most prominent MTB lineage in East Asia, also indicates the possibility that students/teachers could have acquired the infection elsewhere. This could be a study limitation. Also, the reference provided is 25 years old, (i.e. Van Soolingen et al., 1995) and mainly based on IS6110 analysis. Could you please add a newer one reporting the genetic diversity of M. tuberculosis in China?

15. Reference list should be corrected.

Reviewer #4: 1.For case definition part, this manuscript selected TST≥15 mm as cut-off point of “latent TB infection”. However, China national standard for latent TB infection was TST≥15 mm. The author need explain the evidence for this cut-off point. Otherwise it could be a classifications.

Moreover, for a probable case of TB was defined as chest radiography indicative of TB, plus productive cough or haemoptysis for ≥ 2 weeks. Actually it was also not suitable because coughing more than 2 weeks was too long to detect new TB case for a TB outbreak. This definition also need a very good explanation.

2.For results part, the index case had some TB symptoms in her junior school. Is there any data about her junior school? She also could cause the other outbreak in her junior school. These data also had very high value to help us understand this outbreak.

3.In this investigation, the Shaoxing CDC launched TST screening among classmates and teachers. How about roommates?

4.Of the 6 teachers who had latent TB in October 2016, none accepted prophylactic treatment, and none had developed into active TB. This should be discussed and explored potential reasons. In conclusion, is the prophylactic treatment also necessary for the teachers when future outbreaks occur.

5.“The opportunity of exposure varied among the teachers and the students maybe resulted in the LTBI rate was lower among students than among teachers being more frequently exposed to the index case than the majority of the students in the different classrooms having rare interaction with the index case-student. ” However, those teachers as adults also could have more higher latent infection rate due to natural higher background infection rate because China is one of high TB burden country. This manuscript explanation was not reasonable unless additional data could be provided.

6.Usually the prophylactic treatment will continue 3 or 6 months. How about those students treatment? Each student finish whole prophylactic treatment?

6. PLOS authors have the option to publish the peer review history of their article (what does this mean?). If published, this will include your full peer review and any attached files.

Reviewer #1: No

Reviewer #2: No

Reviewer #3: No

Reviewer #4: No

---

## [Author Response · Author response to Decision Letter 0]

23 Nov 2020

Dear Editor,

Thank you for the reviewers’ comments on our manuscript, "An outbreak of pulmonary tuberculosis and follow-up of latent tuberculosis infection in a high school: China, 2016-2019" (PONE-D-20-26099) and the opportunity to address the comments and revise our manuscript. The comments are constructive and valuable, and helped us revise and improve the manuscript. We have studied the comments carefully and have made revisions based on the comments. In this letter, we describe our responses point by point, with reviewer comments in bold font and our response in normal font. The manuscript is provided in track-changes as requested. 

Reviewers' comments:

Reviewer's Responses to Questions

Comments to the Author

1. Is the manuscript technically sound, and do the data support the conclusions?

Reviewer #1: Yes

Reviewer #2: Yes

Reviewer #3: Partly

Reviewer #4: Yes

2. Has the statistical analysis been performed appropriately and rigorously?

Reviewer #1: Yes

Reviewer #2: I Don't Know

Reviewer #3: Yes

Reviewer #4: Yes

3. Have the authors made all data underlying the findings in their manuscript fully available?

Reviewer #1: Yes

Reviewer #2: Yes

Reviewer #3: Yes

Reviewer #4: Yes

4. Is the manuscript presented in an intelligible fashion and written in standard English?

Reviewer #1: Yes

Reviewer #2: No

Reviewer #3: No

Reviewer #4: Yes

5. Review Comments to the Author

Reviewer #1: This well-designed epidemiological study presenting the results of an investigation of a school-based tuberculosis outbreak confirms previous findings in similar settings provided by other authors. An appropriate set of statistics tools well controlled the statistical reliability of epidemiologically correct conclusions, which are strengthened by an important follow-up section of the study. The paper is clearly written.

I support publishing this paper.

Specific questions/comments.

1. Low quality of Fig 2 – please, change the format.

Response:

We appreciate the comment, as this is an important figure. We updated the figure to increase the resolution of the graphic and add labels for complexes which refers to a group of strains with less than three loci variance which are linked with different color. We revised the figure legend to read, “Minimum spanning tree showing clustering by MIRU-VNTR of four M. tuberculosis isolates from this outbreak. Each nodal point represents a cluster with identical genotypes, and the sizes of the nodal points are scaled to the number of strains in the cluster. Red shading indicates the outbreak group, which belong to the Beijing family. “Complex” refers to a group of strains with less than three loci variants that are linked with different colors.”

2. Please, clarify in the text the total number of cultures and the total number of culture-confirmed patients (only 4?) in your study.

Response:

These are important points to address. There were 15 sputum specimens obtained from the index case and the 14 students that shared the same grade with her; 6 were positive, including the index case. Four strains isolated in 2016, including the index case and three classmates were typed with MIRU-VNTR testing.

To address this comment we changed the text to read, “Shaoxing CDC obtained 15 sputum specimens from suspected cases and the index case and performed sputum smears; four were positive, including two specimens positive with one from the index case in 2016 and two specimens positive in subsequent years. M. tuberculosis was isolated from six of the 15 sputum specimens with four strains isolated in 2016 including index case and two strains isolated in subsequent years.”

3. Please mention DST methods and at least phenotypic drug-susceptibility profiles of Mt isolates clustered by MIRU-VNTR.

Response:

Drug-susceptibility testing was performed with liquid rapid drug sensitivity method with the following four tuberculosis drugs: isoniazid, streptomycin, rifampicin and ethambutol. 

To address this comment, we revised the manuscript to say, “Drug-susceptibility testing was performed with the liquid rapid drug sensitivity method with the following four tuberculosis drugs: isoniazid, streptomycin, rifampicin and ethambutol. All four of the isolates were sensitive to these four drugs.”

4. MIRU-VNTR clustered digital profiles should be assigned according to international databases: 

http://www.pasteur-guadeloupe.fr:8081/SITVIT2/query (http://www.pasteur-guadeloupe.fr:8081/SITVIT2/submit.jsp) and https://www.miru-vntrplus.org/MIRU/index.faces

Response:

This is a good point. We identified our cluster as Beijing family based on international database https://www.miru-vntrplus.org/MIRU/index.faces.

The text now states that, “MIRU-VNTR results from the isolates in 2016 demonstrated that these four strains contained the same MIRU-VNTR alleles in 24 loci and minimum spanning tree, indicating that they belonged to the Beijing family based on the international database https://www.miru-vntrplus.org/MIRU/index.faces.”

Additionally, the Figure 2 legend now indicates the Beijing family, stating, “Minimum spanning tree showing clustering by MIRU-VNTR of four M. tuberculosis isolates from this outbreak. Each nodal point represents a cluster with identical genotypes, and the sizes of the nodal points are scaled to the number of strains in the cluster. Red shading indicates the outbreak group, which belong to the Beijing family. “Complex” refers to a group of strains with less than three loci variants that are linked with different colors.”

5. To my mind, the term Homology analysis of isolates does not imply MIRU-VNTR data but rather implies some other tools to compare nucleotide or protein sequences to sequence databases and calculates the statistical significance of matches (https://www.ncbi.nlm.nih.gov/guide/homology/) https://molbiol-tools.ca/Homology.htm Maybe ‘MIRU-VNTR-typing’ will be more relevant?

Response:

This is a good point. We changed “Homology analysis” to “MIRU-VNTR-typing” throughout the manuscript.

6. Line 305 …family, which is the most prominent MTB lineage in East Asia[27], This reference [27] refers to D. van Zoolingen et al. paper of 1995 and definitely should be preserved in the list. However, there is a significant number of quiet recent publications on Beijing strains circulating in China to be referred to additionally.For example: Merker M., Blin C., Mona S. et al. Evolutionary history and global spread of the Mycobacterium tuberculosis Beijing lineage. Nat Genet. 2015;47(3):242-9. https://doi.org/10.1038/ng.3195

Response:

We have included this reference according to the reviewer’s suggestion.

7. Small corrections:

Remove a string of two hyphen-minus characters (--) in VNTR 48--ETRB (column Y in Table, Supporting information).

Response:

We have done so.

Reviewer #2: I could not find any difference between this report and the author's previous report (Outbreak of pulmonary tuberculosis in a Chinese high school, 2009-2010. J Epidemiol 2013;23:307-12), ref. 2 in the bibliography. The settings, findings, and conclusions are identical in both reports. The take-home message of both articles is the same, that is, the importance of early identification of active Tuberculosis cases and the need to treat latent tuberculosis among close contacts of active tuberculosis patients.

Response:

This manuscript describes a more recent outbreak than the outbreak published in 2013. There are two important differences in this outbreak investigation report. First, in this outbreak we had TB isolates for MIRU-VNTR typing and drug resistance analysis. In contrast, the J Epi study mentioned as a limitation, “... without isolates of Mycobacterium tuberculosis we were unable to confirm by genotyping that all TB patients in this outbreak were infected with the same strain.” Second, this outbreak investigation also had a longer follow-up period, which enabled determination of the outcomes from the latent TB cases and the close contacts of the cases. This longer follow-up strengthened our conclusions and recommendations, indicating that more work needs to be done to prevent spread of TB in school-based outbreaks.

The article needs a comprehensive language editing and a more comprehensible data presentation.

Response:

A native English speaker with subject matter familiarity edited the English of the manuscript. He is a senior advisor at China CDC, Lance Rodewald, and is acknowledged in the Acknowledgements section for English editing.

Several issues need the author's attention such as:

1. Why did the authors defined TST>15 as positive rather than TST>5mm, which is the standard definition of TST positivity among recent contacts of infectious tuberculosis cases. The authors should also elaborate regarding the screening procedure (and TST results (>5mm, >10mm, >15mm) distribution, particularly because the citation they provide in this regard is written in Chinese (ref 13), which is not accessible to many "PLOS ONE" readers.

Response:

We used the official definition of latent TB that is used in China: TST＞15 is defined as strong positive infection and recommended to take prophylaxis based on the regulation in references of the “Guideline of China Tuberculosis Control Program” and “Tuberculosis Control and Prevention. Beijing: China Union Medical College press.2004. 

The definition of TST positivity (greater than 15mm) was also the same as was used in the J Epi article the reviewer mentioned above.

To address the reviewer’s point about the distribution of TST results, we have added the following table to the supplementary materials that shows the distribution:

TST Percentage Numerator Denominator

＜5 72.96 362 498

5≤TST＜10 10.44 52 498

10≤TST＜15 7.63 38 498

≥15 9.24 46 498

2. The number of active cases in table 2 does not match (13 and 12 out of 14).

Response:

Point well taken. When we analyzed the risk of TB contraction of probable/confirmed cases based on exposure whether they are in same dormitory with index case, one student-case was excluded from the analysis as a dormitory contact because the student did not live in the dormitory. The student commuted between his home and school every day.

To address the comment, we included a note to Table 2 to explain why this student-case was excluded from the analysis. The note says, “One student-case was not considered as living in the dormitory because the student commuted from home to school every day and did not live in the dormitory.”

3. "Tuberculosis" case definition should include having a positive culture result, and "probable tuberculosis" definition should include a clinical improvement and resolution of chest X-ray findings following treatment.

Response:

This is a very good point, and we appreciate the reviewer identifying our mistake. We did conduct M. tuberculosis culturing during case finding but accidentally did not include it into the confirmed case definition. 

To address this point, we revised the case definition to state, “We defined latent TB infection (LTBI) as a TST induration ≥15mm in diameter in a student or teacher of the high school[13]; probable TB as a chest radiography indicative of TB, plus at least one of the following: productive cough or hemoptysis lasting for ≥2 weeks, or TST≥15 mm; and confirmed TB as two or more positive sputum smears or one positive sputum smear plus a chest radiograph indicative of TB or culture positive with M. tuberculosis.”

5. The article Title is misleading ":China, 2016-2019"

Response:

We revised the title to indicate that this investigation was in a part of China, not the entire country. The title is now, “An outbreak of pulmonary tuberculosis and a follow-up investigation of latent tuberculosis in a high school in an eastern city in China, 2016-2019.” 

Reviewer #3: Review

The manuscript " An outbreak of pulmonary tuberculosis and follow-up of latent tuberculosis infection in a high school: China, 2016-2019” by Yingxin Pei describes outbreak investigation and evaluate a possible nature of this outbreak, along with the source, and transmission route.

This paper describes interesting follow-up study. Authors brought some additional evidence for the importance of post-exposure TB prophylaxis and highlighted possible transmission of M. tuberculosis in school settings.

However, to my opinion, there are few issues that should be addressed.

1. Several studies on the outbreak investigations in schools in China have been published recently such as Pan et al., 2018; Xu et al., 2019; You et al., 2019; Hou et al., 2020; Bao et al., 2019. Authors should clearly emphasize the importance/novelty of the current study and findings.

Response:

The second reviewer made a similar point. We believe that this study adds to the scientific literature with the addition of TB isolation, MIRU-VNTR typing, and drug resistance analysis. The long follow-up period allowed us to determine the outcomes of the latent TB cases and the close contacts of the cases, which reinforces our main conclusions and recommendations. In addition, the father of the index case contracted TB 2 years before the onset of the index case, which suggested this outbreak is an example for TB transmission from the community to the school. 

2. English editing English should be corrected by a native speaker, and all typos should be corrected. Just few examples:

Line 122: please change to “night sweats”. Also: “Teacher” should be in plural?

Line 131: Here and elsewhere. Mycobacterium tuberculosis in italics.

Line 130 Please change to “variable numbers of tandem repeats”

Lines 133-137: please correct punctuation and the usage of “-“ and “—“

Line 138: please correct “typing method was” or “typing methods were” .

Response:

A native English speaker with subject matter familiarity edited the English of the manuscript. He is a senior advisor at China CDC, Lance Rodewald, and is acknowledged in the Acknowledgements section for English editing.

Methods

3. What software was used to compute the minimal spanning tree?

Response:

We used website https://www.miru-vntrplus.org/MIRU/treeBatch.faces to compute the minimal spanning tree.

To address this comment, we now include the sentence, “We used MIRU-VNTRplus to construct the minimal spanning tree (https://www.miru-vntrplus.org/MIRU/treeBatch.faces).”

4. More detailed information about M. tuberculosis isolates obtained in year 2007 should be provided.

Response:

We used 87 strains which were isolated in 2007 during a national survey of drug-resistant tuberculosis in China. To address this comment, we added the following reference in the manuscript “Zhao YL, Xu SF, Wang LX, et al. National Survey of Drug-Resistant Tuberculosis in China. N Engl J Med 2012;366:2161-2170.” 

5. Line 145. What do you mean by “epidemic strains”?

Response:

We appreciate the reviewer pointing this out. We should have used the term “endemic strains.” We have made this change. The strains were isolated in Zhejiang province from 2007 national survey and are the endemic strains in the province where the outbreak occurred. 

6. Some additional data on the population settings of this study should be provided such as persons per room and students per class.

Response:

This is a good point. The revised manuscript now has a sentence that states, “In 2016, the senior high school had eight first-year classes; there were 405 students - 204 males and 201 females. There were approximately 50 students in each class and between 10-12 students in each dormitory.”

Results

7. Please provide P value for the results, as defined in Materials and Methods.

Response:

The Methods section should not have mentioned p-values, since we used 95% confidence intervals instead. We have removed the sentence mentioning p-values from the Methods section. 

8. Lines 250-252. Please rephrase for clarity.

Response:

We have done so. The sentences now say, “Shaoxing CDC obtained 15 sputum specimens from suspected cases and the index case and performed sputum smears; four were positive, including two specimens positive with one from the index case in 2016 and two specimens positive in subsequent years. M. tuberculosis was isolated from six of the 15 sputum specimens with four strains isolated in 2016 including index case and two strains isolated in subsequent years.”

9. Lines 252-255. Did you used any of online resources such as reference databases/analysis tools for the analysis of MIRU-VNTR results such as MIRU-VNTRplus?

Response:

Yes, and we now include this information, stating, “MIRU-VNTR results from the isolates in 2016 demonstrated that these four strains contained the same MIRU-VNTR alleles in 24 loci and minimum spanning tree, indicating that they belong to the Beijing family based on the international database https://www.miru-vntrplus.org/MIRU/index.faces.”

We also included the following reference about the database into our manuscript: Allix-Béguec C, Harmsen D, Weniger T, Supply P, Niemann, S. Evaluation and user-strategy of MIRU-VNTRplus, a multifunctional database for online analysis of genotyping data and phylogenetic identification of Mycobacterium tuberculosis complex isolates. J Clin Microbiol 2008, 46(8):2692-9. doi: 10.1128/JCM.00540-08. Epub 2008 Jun 11. PMID: 18550737; PMCID: PMC2519508.

10. Figure 2 is of insufficient quality (impossible to read) and insufficiently described. How many isolates were used to construct the tree? What genotypes were included? What is the meaning of green color?

Response:

Reviewer one made a similar comment about the figure. We have revised the figure so that it can be more clearly read. We updated the figure legend to provide information about the isolates used to construct the tree, and we recolored the figure. To address the reviewer point we also revised the figure legend, which now says, “Minimum spanning tree showing clustering by MIRU-VNTR of four M. tuberculosis isolates from this outbreak. Each nodal point represents a cluster with identical genotypes, and the sizes of the nodal points are scaled to the number of strains in the cluster. Red shading indicates the outbreak group, which belong to the Beijing family. “Complex” refers to a group of strains with less than three loci variants that are linked with different colors.”

11. The authors stayed (lines 77-81) that China is one of 30 high TB burden countries, and TB outbreaks often occur in institutional settings. Genotyping results are available only for 4 cases. Thus, there is a possibility that all other identified probable, confirmed and LTBI cases did not originated from the index case and/or belonged to this outbreak.

Response:

We believe that the epidemiological investigation provides evidence that the latent TB cases came from the index case, although we agree with the reviewer that certainty is not possible. Fifteen sputum specimens were collected, including from the index case and 14 students who shared the same grade with her, yielding 6 positive results. The four strains isolated in 2016 were typed using MIRU-VNTR. These laboratory data were complemented with an epidemiological investigation due to the links among these 5 cases. Although no further laboratory subtyping was done for subsequent cases, our investigation showed that sharing a classroom as the index case-student was significantly associated with higher risk of probable/confirmed TB (RR = 37.37).The attack rate of the students in the same dormitory as index case was significantly higher than the students in different dormitories (RR=8.71).

12. Line 265. How the ventilation in the classroom and dormitory was assessed? As no evidence of the poor ventilation is provided in the manuscript, this statement is groundless. Please remove or rephrase.

Response: 

We have removed the sentence about ventilation.

13. Line 300-301. This information is quite outdated and not fully correct. First, at present, whole-genome sequencing (WGS) is widely used to determine Mycobacterium tuberculosis relatedness and is known to deliver greater specificity. Second, the finding in study by David et al., 2018 (Reference 25) showed that, in the setting studied, MIRU-VNTR typing and epidemiological risk factors were poorly predictive of close genomic relatedness, assessed by single nucleotide variants. Also, MIRU-VNTR performance varies markedly by lineage. Please rewrite.

Response:

We added one reference and now provide explanation of the usefulness of MIRU-VNTR typing, stating, “As an adjunct to the investigation, we conducted molecular epidemiological analyses. Although whole-genome sequencing (WGS) is increasingly being used to determine Mycobacterium tuberculosis relatedness and is known to deliver greater specificity than MIRU-VNTR, WGS is not always available, and MIRU-VNTR can be used to determine relatedness. MIRU-VNTR has high discriminatory ability and is commonly used to identify clustering of TB cases that have epidemiological links [27-30].”

14. Line 304-307. In my opinion, the fact that Beijing family is the most prominent MTB lineage in East Asia, also indicates the possibility that students/teachers could have acquired the infection elsewhere. This could be a study limitation. Also, the reference provided is 25 years old, (i.e. Van Soolingen et al., 1995) and mainly based on IS6110 analysis. Could you please add a newer one reporting the genetic diversity of M. tuberculosis in China?

Response:

We added two references and updated the narrative to address the reviewer’s good point. The narrative now says, “We found three isolates from classmates of the index case and showed that all of the MIRU-VNTR alleles were identical with the index case’s - all belonging to the Beijing family, which is the most prominent MTB lineage in East Asia [31-32]. The Beijing genotype is the predominant lineage in Zhejiang, and the distribution of Beijing-genotype strains shows geographic diversity [33]. As the Beijing strain is the endemic strain in Zhejiang, this concordance helps explain the outbreak. Genotyping verified and complemented the epidemiological findings that the index case spread TB to her classmates and her roommates through persistent contact due to her delayed diagnosis and lack of initial isolation.”

15. Reference list should be corrected.

Response:

We have done so.

Reviewer #4: 1.For case definition part, this manuscript selected TST≥15 mm as cut-off point of “latent TB infection”. However, China national standard for latent TB infection was TST≥15 mm. The author need explain the evidence for this cut-off point. Otherwise it could be a classifications.

Moreover, for a probable case of TB was defined as chest radiography indicative of TB, plus productive cough or haemoptysis for ≥ 2 weeks. Actually it was also not suitable because coughing more than 2 weeks was too long to detect new TB case for a TB outbreak. This definition also need a very good explanation.

Response:

The second reviewer made a similar point about the TST. We responded with, “We used the official definition of latent TB that is used in China: TST＞15 is defined as strong positive infection and recommended to take prophylaxis based on the regulation in references of the “Guideline of China Tuberculosis Control Program” and “Tuberculosis Control and Prevention. Beijing: China Union Medical College press.2004. 

The definition of TST positivity (greater than 15mm) was also the same as was used in the J Epi article the second reviewer mentioned.

We conducted case searching by actively screening close contacts of the index case who either shared the same grade or same classroom or were family members. Therefore, we believe that cases could be identified in a timely manner since the person will be considered as TB case as long as they demonstrated suspected TB symptoms.

2. For results part, the index case had some TB symptoms in her junior school. Is there any data about her junior school? She also could cause the other outbreak in her junior school. These data also had very high value to help us understand this outbreak.

Response:

We agree with the reviewer that there is value in this information. One classmate of the index case was in the same class as the index case in junior high school. This individual was identified as a TB case during screening in senior high school. No other suspected TB cases were found in the junior high school based on the information provided by teachers in the school. No screening was conducted of other classmates in the junior high school because these students went to different senior high schools after junior high school. None of her junior high school classmates were among the reported TB cases in same administrative area during the 2016-2018 study period.

To address this comment, we added the following text to the manuscript narrative, “The junior high school classmate was identified as a TB case during screening in senior high school. This individual was identified as a TB case during screening in senior high school. No other suspected TB cases were found in the junior high school based on the information provided by teachers. No screening was conducted of other classmates in the junior high school because these students matriculated at different senior high schools. None of her junior high school classmates were among the reported TB cases in same administrative area during the 2016-2018 study period.”

3.In this investigation, the Shaoxing CDC launched TST screening among classmates and teachers. How about roommates?

Response:

This is a good question. All of the dormitory roommates were classmates as well. To address this question, we added this information to the manuscript, stating, “The index case’s dormitory roommates were all screened because they were also the index case’s classmates.”

4. Of the 6 teachers who had latent TB in October 2016, none accepted prophylactic treatment, and none had developed into active TB. This should be discussed and explored potential reasons. In conclusion, is the prophylactic treatment also necessary for the teachers when future outbreaks occur.

Response:

We think so. None of the teachers accepted prophylactic treatment due to concern about side effects of prophylaxis. However, it is recommended to take prophylaxis for latent TB infection during a TB outbreak, even though these teachers with latent TB infection did not develop TB.

5.“The opportunity of exposure varied among the teachers and the students maybe resulted in the LTBI rate was lower among students than among teachers being more frequently exposed to the index case than the majority of the students in the different classrooms having rare interaction with the index case-student. ” However, those teachers as adults also could have more higher latent infection rate due to natural higher background infection rate because China is one of high TB burden country. This manuscript explanation was not reasonable unless additional data could be provided.

Response:

The point is well taken. We revised this sentence to state, “Opportunities for exposure for teachers and students were different, resulting in a lower LTBI rate among students, as the teachers had apparently been more frequently exposed to the index case than the students. Teachers had a higher LTBI rate than the students did, as LTBI rates are known to increase with age. [21]” We added a reference for the sentence: Chen C, Zhu T, Wang Z, Peng H, Kong W, Zhou Y, et al. (2015) High Latent TB Infection Rate and Associated Risk Factors in the Eastern China of Low TB Incidence. PLoS ONE 10(10): e0141511. doi:10.1371/journal.pone.0141511. PMID: 26505997; PMCID: PMC4624631.

6. Usually the prophylactic treatment will continue 3 or 6 months. How about those students treatment? Each student finish whole prophylactic treatment?

Response:

Four students completed the recommended 6-month regimen and none developed TB through the end of the follow-up period. 

We now state, “Shaoxing CDC implemented mandatory isolation and treatment of all probable and confirmed case-students identified during the second round of screening. Free prophylactic treatment was offered to students and teachers found to have latent TB infection. Among those offered prophylaxis, 17.39% (4/23) completed the recommended 6-month regimen, and none developed TB during the follow-up period.”

6. PLOS authors have the option to publish the peer review history of their article (what does this mean?). If published, this will include your full peer review and any attached files.

Do you want your identity to be public for this peer review? For information about this choice, including consent withdrawal, please see our Privacy Policy.

Reviewer #1: No

Reviewer #2: No

Reviewer #3: No

Reviewer #4: No

---

## [Decision Letter · Decision Letter 1]

13 Jan 2021

PONE-D-20-26099R1

An outbreak of pulmonary tuberculosis and a follow-up investigation of latent tuberculosis in a high school in an eastern city in China, 2016-2019

PLOS ONE

Dear Dr. Pei,

Thank you for submitting your manuscript to PLOS ONE. After careful consideration, we feel that it has merit but does not fully meet PLOS ONE’s publication criteria as it currently stands. Therefore, we invite you to submit a revised version of the manuscript that addresses the points raised during the review process.

Please make clarification with regard to the additional comment made by the reviewer: 

There still is a problem about the definition on latent TB infection.  Based on guideline, latent TB infection is that TSTs is over 10  mm.  For preventive therapy, the cut-off point is recommended as TSTs 15 mm for students. The author should clarify it.

We look forward to receiving your revised manuscript.

Kind regards,

Igor Mokrousov, Ph.D., D.Sc.

Academic Editor

PLOS ONE

Reviewers' comments:

Reviewer's Responses to Questions

**Comments to the Author**

1. If the authors have adequately addressed your comments raised in a previous round of review and you feel that this manuscript is now acceptable for publication, you may indicate that here to bypass the “Comments to the Author” section, enter your conflict of interest statement in the “Confidential to Editor” section, and submit your "Accept" recommendation.

Reviewer #4: (No Response)

2. Is the manuscript technically sound, and do the data support the conclusions?

Reviewer #4: Yes

3. Has the statistical analysis been performed appropriately and rigorously? 

Reviewer #4: Yes

4. Have the authors made all data underlying the findings in their manuscript fully available?

Reviewer #4: Yes

5. Is the manuscript presented in an intelligible fashion and written in standard English?

Reviewer #4: Yes

6. Review Comments to the Author

Reviewer #4: 

There still is a problem about the definition on latent TB infection.  Based on guideline, latent TB infection is that TSTs is over 10  mm.  For preventive therapy, the cut-off point is recommended as TSTs 15 mm for students. The author should clarify it.

7. PLOS authors have the option to publish the peer review history of their article (what does this mean?). If published, this will include your full peer review and any attached files.

Reviewer #4: No

---

## [Author Response · Author response to Decision Letter 1]

4 Feb 2021

6. Review Comments to the Author

Reviewer #4: 

There still is a problem about the definition on latent TB infection. Based on guideline, latent TB infection is that TSTs is over 10 mm. For preventive therapy, the cut-off point is recommended as TSTs 15 mm for students. The author should clarify it.

Authors’ Response:

The reviewer makes a good point. We defined latent TB infection (LTBI) as a TST with induration ≥15mm in diameter in a student or teacher of the high school. As the reviewer points out, this cut-off point is the value for initiating prophylaxis in our national TB guidelines. Our use of 15 mm for the definition of LTBI is more restrictive than a 10 mm cut-off, and will identify fewer individuals in need of prophylaxis. Since we used national guidelines for definitions, this is a limitation of our study. We now recognize this as a limitation in the manuscript, and we state in the limitations section, “Third, as per national guidelines for prophylaxis, we defined latent TB infection (LTBI) as TST induration ≥15mm in diameter. This cut-off is more conservative than a ≥10mm cut-off, and will result in fewer LTBI cases being shown to progress to TB than would have been shown had a 10 mm cut-off been used.”

---

## [Editor Report · Decision Letter 2]

10 Feb 2021

An outbreak of pulmonary tuberculosis and a follow-up investigation of latent tuberculosis in a high school in an eastern city in China, 2016-2019

PONE-D-20-26099R2

Dear Dr. Pei,

We’re pleased to inform you that your manuscript has been judged scientifically suitable for publication and will be formally accepted for publication once it meets all outstanding technical requirements.

Kind regards,

Igor Mokrousov, Ph.D., D.Sc.

Academic Editor

PLOS ONE
---

## [Editor Report · Acceptance letter]

15 Feb 2021

PONE-D-20-26099R2 

An outbreak of pulmonary tuberculosis and a follow-up investigation of latent tuberculosis in a high school in an eastern city in China, 2016-2019 

Dear Dr. Pei:

I'm pleased to inform you that your manuscript has been deemed suitable for publication in PLOS ONE. Congratulations! Your manuscript is now with our production department. 

Kind regards, 

on behalf of

Dr Igor Mokrousov 

Academic Editor

PLOS ONE